# Development and Optimization of a Real-Time Monitoring System of Small-Scale Multi-Purpose Juice Extractor

**DOI:** 10.3390/foods14020227

**Published:** 2025-01-13

**Authors:** Tae-Hyeon Kim, Jae-Min Jung, Wang-Hee Lee

**Affiliations:** 1Department of Smart Agriculture Systems, Chungnam National University, Daejeon 34134, Republic of Korea; fhfh1515@naver.com; 2Department of Biosystems Machinery Engineering, Chungnam National University, Daejeon 34134, Republic of Korea; jaemin0902@gmail.com

**Keywords:** juice extractor, juicing process, optimization, postharvest food processing, smart monitoring

## Abstract

According to the concept of smart postharvest management, an information and communication technology sensor–based monitoring system is required in the juicing process to reduce losses and improve process efficiency. Such technologies are considered economically burdensome and technically challenging for small-scale enterprises to adopt. From this perspective, this study aimed to develop a smart monitoring system for the juicing processes in small-scale enterprises and to identify the optimal operating conditions based on the monitoring data. The system developed is equipped with two weight sensors attached to the twin-screw juice extractor, allowing for the automatic measurement of the weight of the raw material and the resulting juice product. The measured data are automatically transmitted and stored on a computer. Additionally, the system was designed to remotely control the speeds of the juicing and feeding screws, which are the primary controlling factors of the twin-screw juicer. Juice yield and processing time were optimized using carrots and pears. The optimal juicing and feeding speeds for pear yield were found to be 167.4 rpm and 1557 rpm, respectively; carrots achieved an optimal yield at a juicing speed of 502.2 rpm and feeding speed of 1211 rpm. In contrast, the processing time was minimized at juicing–feeding speeds of 6–6 and 7–5 for pears and carrots, respectively. Consequently, it was challenging to determine the optimal conditions for simultaneously optimizing the yield and processing time. This also suggests that the juicing process is affected by the properties of the fruits and vegetables being processed. By developing a system capable of accumulating the data necessary for the digitization of postharvest management and food processing, this research offers a valuable platform for the smart monitoring and optimization of the juicing process.

## 1. Introduction

The consumption of processed foods is evolving in response to changing dietary patterns, with a growing emphasis on healthier ingredients [1]. This trend has fueled interest in nutritious foods, favoring nutrient-dense products achievable via juicing and concentration [2,3,4,5]. Consequently, the consumption of juice products has steadily increased, prompting their consideration as a valuable addition to the future of food [6,7,8]. Smart postharvest management systems that optimize the quality and shelf life of processed agricultural products, such as juices, are also gaining attention. Smart postharvest management integrates advanced technologies such as information and communication technology, big data analytics, and artificial intelligence to develop data-driven systems for efficiently managing agricultural products after harvest [9,10]. These systems, incorporating remote sensing-based real-time monitoring and postharvest processing optimization, improve efficiency and reduce nutrient loss, meeting the demand for healthier and more sustainable food options [9,10].

Juice extractors are one of the most widely used traditional food processing machines in food engineering. Different types of juice extractors, such as hydraulic and screw juice extractors, are used depending on the raw material and the desired product characteristics. Quality and yield are paramount factors in juice extraction and are determined by the mechanical performance of the extractor. This highlights the need for optimization via monitoring [11,12]. There are several studies on the optimization of the juice extraction process based on experimental data. Examples include identifying optimal process variables (blending speed, extraction time, and ripening stage) for mechanically juicing bananas [13] and optimizing conditions to enhance the quality attributes of carrot juice [14]. However, compared to the abundance of studies on juice quality and juice extractor performance [15,16,17,18,19], studies on the optimization of juice processing with real-time monitoring remain limited. Despite the increasing focus on maximizing yield and profits in smart food processing, this remains a challenge [20,21,22,23].

The development of remote sensing provides exciting opportunities for real-time data acquisition and food processing control [24,25]. Even though remote sensing can reduce production costs and increase profits by optimizing food processing, current juice extraction monitoring primarily focuses on hygiene, suggesting that optimization and real-time monitoring for increasing economic benefits are still underdeveloped [26]. Unlike large corporations with sufficient resources and advanced technical capabilities, small-scale enterprises often face delays in acquiring and implementing monitoring system technologies. Furthermore, small-scale food processing enterprises often lack the resources for customized monitoring systems for juice extraction processes. For this reason, despite the emphasis on smart and digital agriculture for achieving sustainable post-harvest processing, small-scale juice extraction continues to rely on traditional experience-based methods. This underscores the potential to enhance cost and quality efficiency while producing consumer-driven products through effective juice extraction monitoring and the identification of optimal operating points. Therefore, the present study aims to develop a highly effective real-time monitoring system for screw-type juice extractors commonly used in small-scale enterprises. Our goal is to identify the optimal operating conditions for maximizing the yield and minimizing the processing time and provide a system that can be readily implemented in small-scale enterprises for effective process management based on the comprehensive monitoring data sets, leading to profits. This work not only lays the foundation for real-time juice monitoring but also paves the way for future optimization efforts.

## 2. Materials and Methods

### 2.1. System Development

#### 2.1.1. Juicing Machine Description

A small twin-screw juice extractor (BC-G200; Sejong Hitech Co., Ltd., Cheongju, Republic of Korea) was selected for this study because of its popularity and practicality in South Korea. The selected small twin-screw juice extractor measures 1100 mm in length, 500 mm in width, 1350 mm in height, and weighs 200 kg. Its primary components include a hopper, feeding screw, juicing screw, juicing net, juice/sludge outlet, and control panel. The extraction process involves feeding raw materials into the hopper, where they are crushed by the feeding screw. The crushed material is then transported to the juicing screw, which squeezes the material against the juicing net. The feeding screw was driven by a 1/2HP 3-phase motor and the juicing screw by a 2HP 3-phase motor.

#### 2.1.2. Designing the Monitoring System

To design a monitoring system for juice extraction that measures the real-time input/output weight and controls the screw speed with wireless storage of data, two main components were employed: a load cell and a programmable logic controller (PLC) [27]. Two beam load cells, each with dimensions of 530 mm × 390 mm (width × length) and a maximum load capacity of 100 kg (SB-100K, A&D KOREA Ltd., Seoul, Republic of Korea) were used to measure the amount of input (raw material) and output (juice). These load cells were placed in front of the handle on the cart platform and below the juice outlet. The cart platform was designed to measure the real-time input into the existing juicer by comparing the weight of the entire system with the weight of the juicer. The cart platform had a width of 1400 mm, a length of 550 mm, and a maximum loading capacity of 500 kg (Figure 1a). The screw speed was controlled using a PLC (XBC-DR28U; LS ELECTRIC Co., Ltd., Anyang, Republic of Korea) installed on the back of the control panel. The input and output signals of the screw motor were transmitted via the PLC to a personal computer (PC) via a router (WL-WN572HG3, WAVLINK Technology Ltd., Shenzhen, China) (Figure 1b).

#### 2.1.3. Operating the Monitoring System

The monitoring system controlled the screw speeds of the juicing and feeding motors. Each motor offered 10 screw speed options, ranging from 167.4 to 1730 rpm for juicing and 173 to 1674 rpm for feeding. During juice extraction, the system collected monitoring data every 0.1 s and transmitted it to a PC via wireless fidelity (Wi-Fi) using the PLC. These data were then stored on the PC as comma-separated values file with 11 fields: processing time, starting weight, current weight, juice weight, yield, expected yield, current weight increase rate, juice weight decrease rate, additional weight, juicing speed, and feeding speed. Simultaneously, the received data were projected onto a graphical user interface (GUI) program. This GUI program comprised a display panel of real-time measurement data, a screw speed adjustment section, a system on/off control panel, and an alarm confirmation section. The field-value yield of the received data was calculated using formula (Equation (1)):(1)$$Yield\left(\%\right)=\frac{Initial\;weight\;-\;Current\;weight}{Juice\;weight}.$$

### 2.2. Experiment

#### 2.2.1. Raw Materials

Pears and carrots were chosen as the juicing raw materials because of their popularity in juices and market affordability. To reflect the differences in the juice extraction due to hardness, we selected these two items, which are widely produced and easily available in Korea, and are also among the most popular choices in the juice extraction process. The selected pears were *Pyrus pyrifolia*, primarily cultivated in Asia, whereas the carrots were *Daucus carota* L, cultivated globally [28,29]. A total of 1250 kg of pears and carrots were used in this experiment. The pears were purchased from a farm in Cheonan, South Korea, and the carrots were purchased from the Noeun Agricultural and Fisheries Market in Daejeon, South Korea. Upon receipt, each raw material was cleaned and washed on the same day before the experiment. The raw materials were then divided into 25 groups of 50 kg each, following the experimental design.

#### 2.2.2. Experimental Design

The juice extraction experiment was conducted by varying the screw speed of the juice extractor, which subsequently affected the yield and processing time. The stages of motor speed for adjusting the screw speed of the juice extractor ranged from 0 (stop) to 10 (maximum speed). While it was technically possible to adjust all stages, we determined that the difference in speed between consecutive stages (average of 170.2 rpm) was not significant enough to affect the extraction results. Therefore, the screw speed was divided into five stages. Five screw speeds (1, 3, 5, 7, and 9) were applied for both juicing and feeding, resulting in 25 experiments with different speed combinations. The screw speeds in the juicing motor corresponded to the following rpm levels: 1 (167.4 rpm), 3 (502.2 rpm), 5 (837 rpm), 7 (1171.8 rpm), and 9 (1506.6 rpm). The screw speeds in the feeding motor corresponded to the following levels: 1 (173 rpm), 3 (519 rpm), 5 (865 rpm), 7 (1211 rpm), and 9 (1557 rpm). As previously mentioned, each raw material, weighing 50 kg, was used in 25 experiments to measure the yield and processing time.

#### 2.2.3. Experimental Operation and Data Acquisition

The juicing process for each experimental group followed a specific sequence: raw material washing, weighing and input, juicing, and juice extractor cleaning (Figure 2). Each group of 50 kg of the raw materials was washed, prepared, and input into the hopper in three separate batches because of the hopper’s capacity limit. The monitoring system was then used to control the juicer at the designated juicing and feeding speeds. To eliminate experimental errors caused by initial screw rotation before juice extraction in the first 2–3 min, a pre-extraction of 5–6 kg of raw materials was performed before starting the experiment. When the remaining raw material in the hopper was less than 1 kg, additional raw material (16–17 kg) was added. Finally, the juice extraction was stopped when the weight of the raw material reached 0. During the experiment, real-time juicing data were collected every 0.1 s, transferred, and saved on the PC immediately after the raw material input. This resulted in a dataset of real-time measurements from the 25 experimental groups, containing 11 fields for each extraction. These fields included key variables such as the yield, processing time, juice weight, input amounts, and juicing and feeding speeds.

### 2.3. Data Analysis and Optimization

#### 2.3.1. Data Preprocessing

Raw data directly stored from the juice extractor monitoring system were preprocessed for further optimization without overfitting and bias [30]. As juicing did not begin immediately, the initial data points fluctuated between zero and the actual juice weight readings. Therefore, all data values recorded before the first non-zero weight were removed. Outliers were identified and removed based on the standard deviation of the residuals from a linear regression of juice amount against processing time. The three-sigma rule was modified to a four-sigma rule to detect outliers. Compared to the three-sigma rule, the four-sigma rule offered superior performance in identifying actual outliers as documented in studies by Ling [31] and Pukelsheim [32]. To facilitate optimization, data from both pear and carrot experiments were combined into a single dataset incorporating all the experimental results. The final datasets for optimization were composed of the processing time, juicing weight, yield, juicing speed, and feeding speed.

#### 2.3.2. Optimization Using Response Surface Methodology (RSM)

RSM was employed to optimize the juicing process for the yield and processing time as a function of the juicing and feeding speeds, controlled by screws in the extractor [33,34]. The preprocessing results for the pears and carrots were formulated using RSM with a second-order polynomial (Equation (2)):(2)$$Y=\beta_{0}+\beta_{1}X_{1}+\beta_{11}X_{1}^{2}+\beta_{2}X_{2}+\beta_{22}X_{2}^{2}+\beta_{12}X_{1}X_{2},$$
where β is a model parameter determined based on the experimental data, and X_i_ is an independent variable, which are juicing and feeding speed in this study.

The stationary point, also referred to as the critical point, was investigated based on the eigenvalues that defined the types of stationary points: minimum, maximum, and saddle points. When all eigenvalues were negative, the stationary point corresponded to the maximum value, whereas it corresponded to the minimum value with all positive eigenvalues. If the eigenvalues alternate between positive and negative, the stationary point is known as the saddle point. Additionally, a lack-of-fit test was performed to evaluate the suitability of the model structure, and the results were compared to the statistical significance level of 0.05. To perform RSM, the ‘rsm’ package in R software was employed [35,36]. Statistical significance was determined at *p*-value < 0.05 for the whole model and each term, whereas the lack-of-fit test with a *p*-value greater than 0.05 indicated that the model structure was suitable.

## 3. Results

### 3.1. Juice Extraction Monitoring and Data

The developed monitoring system can remotely control the juicing and feeding screws and obtain real-time data, such as the juice volume, processing time, and yield. During the juicing process, the monitoring data are displayed on the GUI, and alarms are triggered in the case of juicer malfunctions such as system errors or exceeding capacity limits.

The average yields for pears and carrots were approximately 78% and 34%, respectively, with a much lower yield for carrots (Table 1). The juicing of carrots was only successful with five out of twenty-five speed combinations: 3–3, 3–5, 3–7, 5–3, and 7–3 (juicing speed-feeding speed). A maximum yield of 84% was observed for pears at juicing and feeding speeds of 1 and 7, respectively. The carrot juice extraction showed a maximum yield of 37% at juicing and feeding speeds of 3 and 7, respectively. The yield variation for pears was larger than that of carrots due to the wide range of speed operations, suggesting that optimization would be more effective for pear juice extraction. The processing time for pears was 768.5 s, ranging from 281.9 to 2163.7 s depending on the juicing–feeding speeds; carrot juicing required more time, with an average processing time of 1008.16 s. The shortest processing time for pears was observed at juicing-feeding speeds of 7–9, indicating a shorter time with high speeds of both juicing and feeding, as expected. However, the shortest processing time for carrot juice extraction was at juicing–feeding speeds of 7–3, indicating that the processing time was influenced more by the juicing speed than the feeding speed.

### 3.2. Optimization of Yield and Processing Time

The coefficient of determination (R^2^) was the lowest for the processing time model for carrot juice extraction (R^2^ = 0.194), whereas the highest R^2^ of 0.851 was observed for the carrot yield evaluation (Table 2). All linear, quadratic, and cross-product terms were substantial, indicating that both the juicing and feeding speeds, as well as their combination affected the processing time and yield. The lack-of-fit test yielded a *p*-value greater than 0.05, indicating that the developed RSM model structure was suitable for capturing the data response. To visually present the optimization results based on RSM, the changes in yield and processing time with respect to juicing and feeding speeds were illustrated using contour maps and 3D plots of the response surface (Figure 3 and Figure 4). The stationary points for the RSM were minimum for all cases. For processing time, the lowest values were found for pears at juicing and feeding speeds of 5.992 (1003.1 rpm) and 6.447 (1115.3 rpm), respectively. Similarly, the processing time for carrots was minimized at 7.476 (1251.5 rpm) for the juicing speed and 4.661 (806.4 rpm) for the feeding speed. For the yield, the lowest values were found for pears at juicing and feeding speeds of 7.351 (1230.6 rpm) and 0.333 (0 rpm), respectively, whereas for carrots, the lowest values were observed at juicing and feeding speeds of 4.661 (780.3 rpm) and 3.459 (598.4 rpm), respectively. However, contrary to expectations, the response surface for the yield was minimal, necessitating further statistical analysis of the monitoring data to determine the optimal operating conditions of the juice extractor. To determine the maximum value from the minimum value, the last point of the curve with the steepest slope within the limited screw speed range was selected as the maximum yield. Therefore, for pears and carrots, the maximum yields were found at juicing–feeding speeds of 1–9 and 3–7, respectively. Additionally, by deriving a value that can maximize the yield and minimize the processing time, 82.38% yield and 151.84 s processing time for pears (juicing–feeding speeds of 1–7) and 36.98% yield and 1563.29 s processing time for carrots (juicing–feeding speeds of 3–7) were obtained (Table 3).

### 3.3. Changes in Optimal Yield and Processing Time by Screw Speeds

We investigated changes in the optimal yield and time with variations in the speed of juicing and feeding at their optimal levels (Figure 5). Owing to discrepancies in the dimensions of the time and yield, we initially standardized both datasets into Z-scores. According to the results, the processing time exhibited a higher variation than the yield in pear juicing, indicating that the screw speed exerted a greater impact on the processing time than on the yield. Additionally, based on the eigenvectors observed in the RSM model, the juicing speed was a more important factor than the feeding speed, whereas the feeding speed had a greater influence on the processing time. However, although the yield was highly dependent on the juicing speed, the difference in the impact between the feeding and juicing speeds on the processing time was smaller than their impact on the yield. In contrast, for carrot juicing, the processing time and yield were similarly affected by the screw speed, suggesting that both the yield and processing time are influenced by the physical properties of the agricultural products. Interestingly, varying the juicing speeds at a fixed feeding speed resulted in similar changes in the yield and processing time for both pears and carrots. However, when the feeding speed was varied at a fixed juicing speed, different patterns were observed for the yield and processing time. This indicates that the feeding speed is more influenced by the physical properties of the fruits and vegetables being juiced. This can be analyzed using eigenvectors in the RSM model, showing that the feeding speed predominantly affects the yield and processing time of carrots. This suggests that the high potential for carrot sludge buildup, owing to their hardness, can cause a bottleneck in the juicing process, indicating a need for an adjustment in the feeding speed.

## 4. Discussion

Agricultural systems are continually evolving to tackle challenges such as population growth, food security, and climate change [37,38,39,40]. Among these advancements, agricultural monitoring systems in processing are essential for achieving efficient production and ensuring consistent product quality [41,42]. Currently, traditional processing methods lack effective monitoring systems to optimize productivity; instead, the focus has primarily been on hygiene, with hazard analysis critical control points (HACCP) used to manage food safety [16]. As a result, there is an increasing demand for research into systems that can monitor agricultural and food processing under optimal operating conditions.

In the present study, a real-time juice monitoring system was developed to effectively manage the fruit juicing process and identify the optimal operational conditions for the machine, which holds promise for establishing digital post-harvesting systems in small and medium industries. Alongside quality maintenance, the yield and processing time are critical determinants of efficiency and productivity in agricultural processing. The juice quality is directly influenced by the quality of the raw materials, which is contingent upon the selection of the agricultural products [43]. Therefore, monitoring and controlling mechanical factors are essential to enhance the consistency and efficiency of the process. Overall equipment effectiveness (OEE) is a metric calculated as the product of the availability, performance, and quality. A high OEE necessitates the efficient operation of agricultural processing equipment, which in juicing translates to optimizing the yield and processing time [44,45,46,47]. The present study focused on the juicing and feeding screw speeds that are easily adjustable mechanical factors and significantly impact the yield and processing time. The RSM analysis revealed a significant variability in the yield and processing time with changes in the two screw speeds. However, identifying the optimal operating conditions that simultaneously maximize the yield and minimize processing time remains challenging. Our results indicate that pear and carrot juicing yields can be improved with high feeding and low juicing speeds, while the fastest processing times are achieved with intermediate speeds. These findings highlight the complexity of balancing the yield and processing time, as optimizing one factor often necessitates compromising the other. A limitation of RSM is its reliance on relatively small datasets and its inability to fully capture nonlinear interactions between variables. Although the lack-of-fit test indicated that the model structure was adequate for capturing the yield and processing time responses for each material, the relatively large standard errors of the parameters in the RSM model, particularly for carrots, suggest the need for improvements in the data volume and quality. To improve the effectiveness of the RSM in future studies, incorporating a hybrid approach that combines the RSM with AI algorithms could help address these limitations. For instance, leveraging machine learning models to refine and validate RSM predictions or using advanced techniques like Gaussian process regression could enhance the precision of optimization outcomes. Moreover, as data are continuously accumulated, the integration of these methods could provide deeper insights and more robust decision-making frameworks, especially for complex systems.

As mentioned above, in addition to mechanical factors, the physical properties of agricultural products significantly affect the juicing process [48,49]. This is reflected in the relatively large standard error of the RSM model parameters for carrots compared to pears. The issue may stem from insufficient data, as the juicing of carrots was conducted within a relatively narrow screw speed range compared to pears. However, it seems more likely that this problem is due to the specific physical characteristics of the carrot compared to the pear. The hardness of carrots was higher than the torque capacity of the motor in the small twin-screw juice extractor at speed levels below 2 and above 7, leading to unsuccessful juicing events. Fruits and vegetables with high fiber content generally exhibit increased hardness [50]. Pears, which contain a significant amount of pectin and soluble dietary fiber, have a relatively lower hardness compared to carrots. In contrast, carrots, as a root vegetable, possess a firmer texture due to their high concentration of insoluble dietary fibers, such as cellulose and hemicellulose [51,52,53]. Additionally, carrots have densely packed cells, resulting in narrow fiber spacing and increased hardness [54,55]. This physical characteristic can lead to insufficient pushing force during juicing, potentially causing the juicing net to become clogged unless adequate torque is applied. Considering the physical properties of raw materials, the mesh size of the juicing net, which covers the juicing screw and directly processes the cut produce, may influence the juicing yield. However, our additional experiments with different mesh sizes (0.5 mm, 0.8 mm, and 1.0 mm) showed minimal changes in yield, consistent with findings from previous studies [56,57]. This suggests that filtration before juicing is essential to remove sludge and improve juicing efficiency, especially for fruits and vegetables with low moisture content and hard textures, such as carrots. Furthermore, in most juicing processes, the raw materials are pre-selected, and the quality of these materials significantly impacts the juicing outcome. If the condition of the raw materials can be evaluated prior to juicing, it could enhance the effectiveness of real-time monitoring systems. Although an increase in temperature due to screw rotation could potentially affect the juicing process, particularly the product quality, it was found that the temperature rise is not significant for this type of juice extractor. The motor generating heat and the path through which the raw material is juiced are separated by at least 20 cm, which effectively minimizes any temperature increase. Moreover, even when heat transfer occurs, the brief contact time during processing (operating at a capacity of 300 kg/h) is insufficient to adversely impact the product. The absence of any negative effect from the heat generation on the product quality was confirmed by the consistent sugar content measured during the juicing process, as demonstrated in previous studies [58,59]. In conclusion, it can be stated that in twin-screw juicers, controlling mechanical factors based on the physical properties of agricultural products is essential for effectively monitoring and optimizing the juicing process.

## 5. Conclusions

In this study, a monitoring system tailored for a juice extraction process in small industries was developed, and both the yield and processing time were optimized using this system. Despite limitations such as using a single type of juicer and a limited variety of raw materials, this study introduces systems and methods for enhancing the management of agricultural product processing, particularly in an era that emphasizes digital postharvest management. Additionally, by optimizing the juice extraction process based on adjustable mechanical factors, valuable postharvest management data were obtained, leading to the development of a system that can be utilized by small-scale enterprises. Future research could focus on expanding this monitoring system beyond monitoring the yield and processing time to include quality factors. While we believe that addressing the issues stemming from the robustness of RSM is still necessary, it appears that these challenges can be resolved by increasing the volume of data through the accumulation of continuous monitoring data and applying AI-based decision-making methods. For instance, if continuous data accumulation is achieved through studies like this, it could enable the application of AI-based algorithms for big data analysis, facilitating real-time monitoring and optimization—capabilities that are challenging to achieve with offline static optimization methods. This expansion could focus on integrating pre-juicing and post-juicing processes to enable the selection of raw materials based on their quality, thereby advancing the digitization of the entire juicing process. Finally, we would like to emphasize that the system needs to be tested in real-world conditions to evaluate its effectiveness in monitoring the juicing process and its resource efficiency.

## Figures and Tables

**Figure 1 foods-14-00227-f001:**
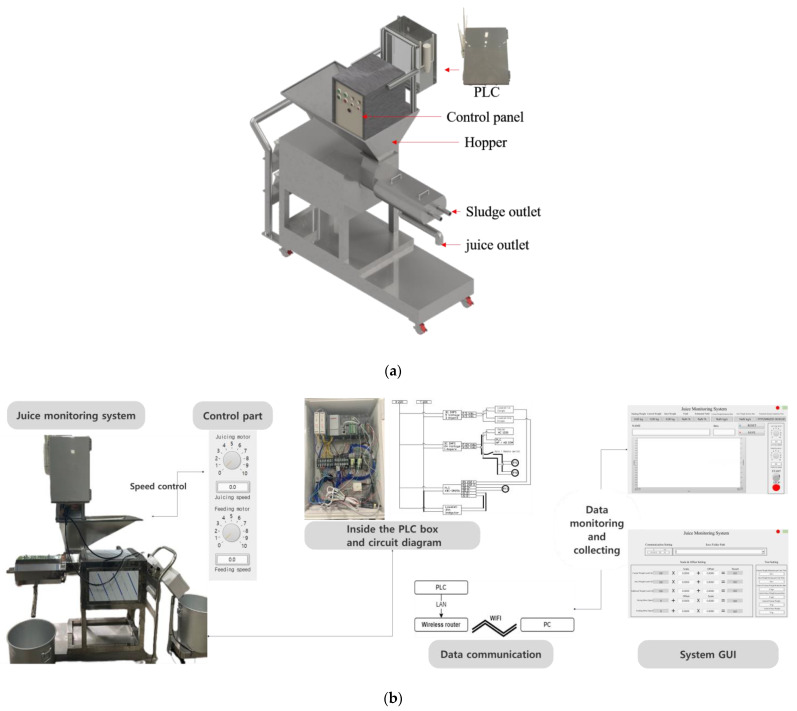
(**a**) Three-dimensional (3D) drawing of juice monitoring system. (**b**) Data flow of juice monitoring system.

**Figure 2 foods-14-00227-f002:**
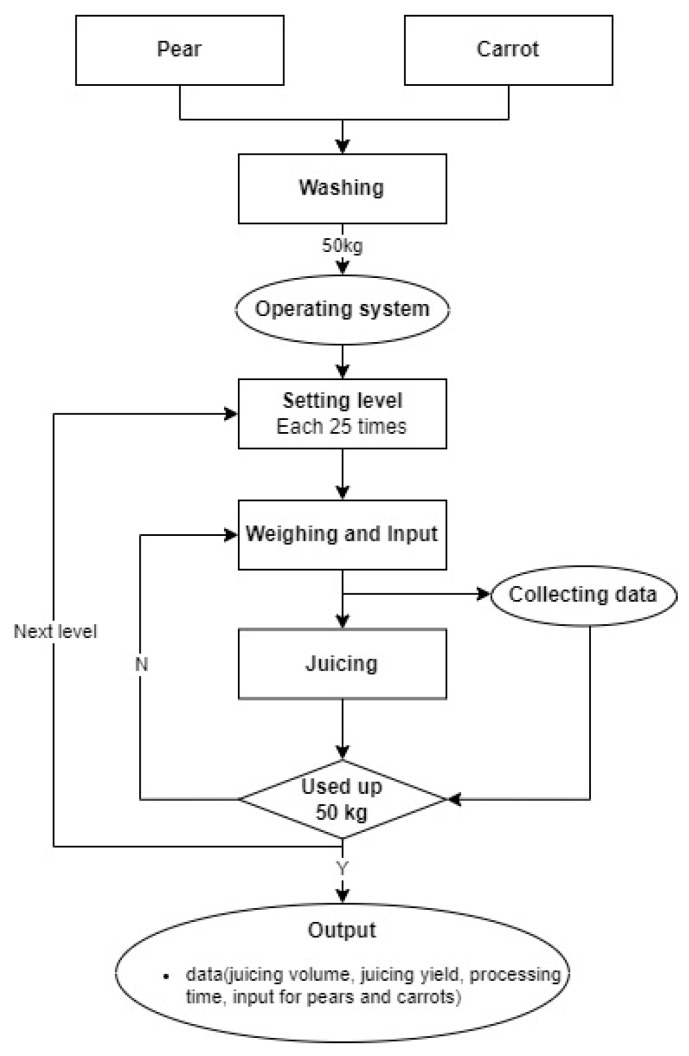
Experimental operation of juice monitoring system.

**Figure 3 foods-14-00227-f003:**
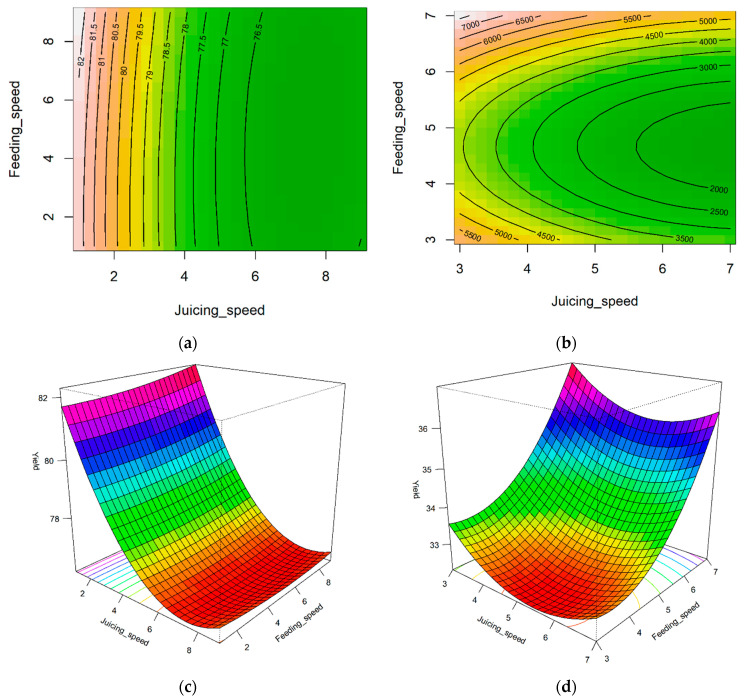
Contour graphs and 3D graphs of pear and carrot yield data: (**a**) Contour graphs of pear yield. ((**a**) indicates that red corresponds to high yield, while green corresponds to low yield.) (**b**) Contour graphs of carrot yield. ((**b**) indicates that red corresponds to a long processing time, while green corresponds to a short one.) (**c**) Three-dimensional graph of pear yield. ((**c**) indicates that purple corresponds to high yield, while red corresponds to low yield.) (**d**) Three-dimensional graph of carrot yield. ((**d**) indicates that purple corresponds to a long processing time, while red corresponds to a short one.).

**Figure 4 foods-14-00227-f004:**
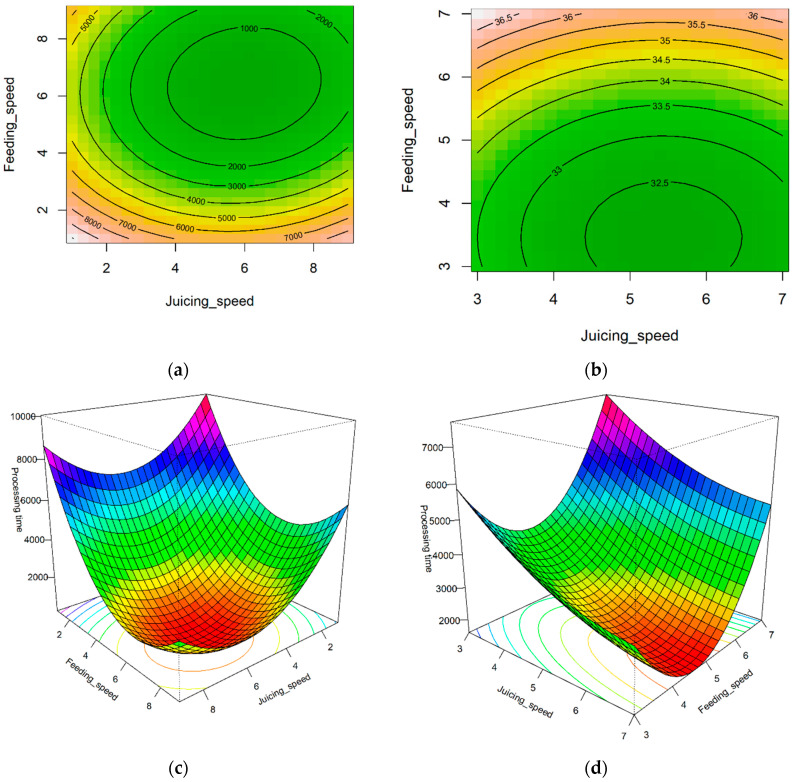
Contour graphs and 3D graphs of pear and carrot yield processing time data: (**a**) Contour graph of pear processing time. ((**a**) indicates that red corresponds to high yield, while green corresponds to low yield.) (**b**) Contour graph of carrot processing time. ((**b**) indicates that red corresponds to a long processing time, while green corresponds to a short one.) (**c**) Three-dimensional graph of pear processing time. ((**c**) indicates that purple corresponds to high yield, while red corresponds to low yield.) (**d**) Three-dimensional graph of carrot processing time. ((**d**) indicates that purple corresponds to a long processing time, while red corresponds to a short one.).

**Figure 5 foods-14-00227-f005:**
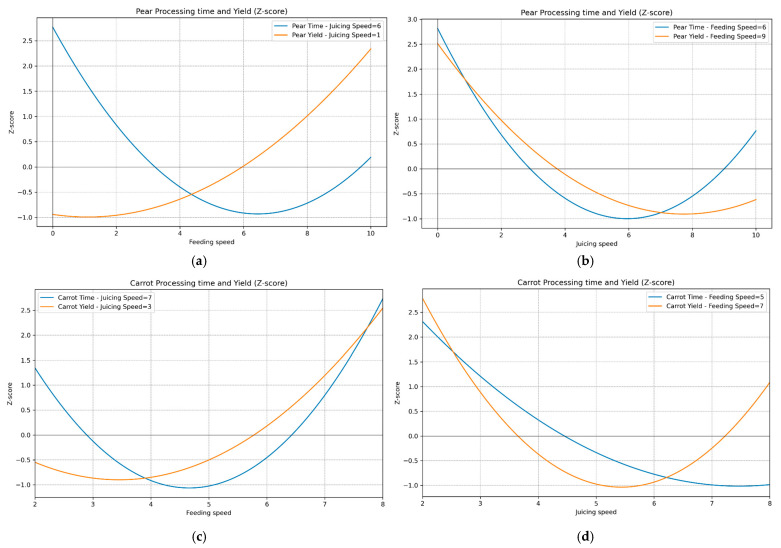
Variations in the yield and processing time standardized to the Z-score with fixed screw speeds at their optimum. (**a**) Yield and processing time of pear at the optimal juicing speed. (**b**) Yield and processing time of pear at the optimal feeding speed. (**c**) Yield and processing time of carrot at the optimal feeding speed. (**d**) Yield and processing time of carrot at the optimal juicing speed.

**Table 1 foods-14-00227-t001:** Descriptive statistics results.

Materials	Factor	Descriptive Statistics	Value	Juicing Speed	Feeding Speed
Pear	Yield (%)	Maximum	83.82	1	7
		Median	77.38	7	5
		Minimum	75.36	9	7
		Mean	78.10	-	-
		Standard deviation	2.23	-	-
	Time(s)	Maximum	2163.7	1	1
		Median	570.1	3	5
		Minimum	281.9	7	9
		Mean	768.53	-	-
		Standard deviation	502.15	-	-
Carrot	Yield(%)	Maximum	36.98	3	7
		Median	33.56	3	3
		Minimum	32.38	5	3
		Mean	33.98	-	-
		Standard deviation	1.62	-	-
	Time(s)	Maximum	1530.8	3	7
		Median	828.1	5	3
		Minimum	681.1	7	3
		Mean	1008.16	-	-
		Standard deviation	307.62	-	-

**Table 2 foods-14-00227-t002:** Result of RSM model.

Materials	Factor	Parameter	Estimate	Standard Error	t Value	Pr > |t|
Pear	Time	Intercept	14,180.912	36.686	386.553	<0.001
		X_1_ *	−1850.801	13.591	−136.177	<0.001
		X_2_ **	−2631.888	13.757	−191.309	<0.001
		X_1_ X_1_	170.419	1.277	133.501	<0.001
		X_1_ X_2_	−29.700	0.989	−30.039	<0.001
		X_2_ X_2_	217.913	1.326	164.340	<0.001
	Yield	Intercept	83.571	0.01	8498.348	<0.001
		X_1_	−1.976	0.004	−542.267	<0.001
		X_2_	−0.011	0.0003	−4.059	<0.001
		X_1_ X_1_	0.134	0.0003	391.039	<0.001
		X_1_ X_2_	−0.011	0.0003	−42.483	<0.001
		X_2_ X_2_	0.010	0.0004	27.500	<0.001
Carrot	Time	Intercept	22,807.938	371.212	61.442	<0.001
		X_1_	−1861.500	109.672	−16.973	<0.001
		X_2_	−6130.000	105.581	−58.060	<0.001
		X_1_ X_1_	124.50	11.087	11.229	<0.001
		X_2_ X_2_	657.562	10.444	62.961	<0.001
	Yield	Intercept	41.668	0.205	266.346	<0.001
		X_1_	−2.23	0.058	−144.824	<0.001
		X_2_	−1.92	0.061	−17.262	<0.001
		X_1_ X_1_	0.205	0.006	131.153	<0.001
		X_2_ X_2_	0.278	0.006	16.934	<0.001

* Juicing speed; ** Feeding speed.

**Table 3 foods-14-00227-t003:** RSM results of optimal operating conditions.

**Materials**	**Factor**	**Juicing Speed (rpm)**	**Feeding Speed (rpm)**
Pear	Time	5.992 (1003.1)	6.447 (1115.3)
	Yield	1 (167.4)	9 (1506.6)
Carrot	Time	7.476 (1251.5)	4.661 (806.4)
	Yield	3 (502.2)	7 (1211)

## Data Availability

The original contributions presented in the study are included in the article, further inquiries can be directed to the corresponding author.

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
