# Peer review of "Development and Optimization of a Real-Time Monitoring System of Small-Scale Multi-Purpose Juice Extractor"

_foods, 2025, doi:10.3390/foods14020227_

Round 1
Reviewer 1 Report (Previous Reviewer 1)
Comments and Suggestions for Authors
Accept in present form
Reviewer 2 Report (Previous Reviewer 2)
Comments and Suggestions for Authors
I acknowledge the author's reply and agree to accept it.
This manuscript is a resubmission of an earlier submission. The following is a list of the peer review reports and author responses from that submission.
Round 1
Reviewer 1 Report
Comments and Suggestions for Authors
A real-time monitoring system of small-scale multi-purpose juice extractor was developed and optimized in the study, which used in small- or medium-sized enterprises. It’s interesting. It need major revision. The comments as follows:
1. What is the level of existing system of juice extractor, and what are the advantages of this system studied in this paper?
2. Why were pears and carrots chosen for the study?
3. How to design the experiment? Why choose the screw speed 1,3,5,7 and 9?
4. Please make detail describe some figures.such as figure3,4
5. Why choose RSM model? Why not other intelligent methods?
6. How useful is this system? Is it actually evaluated in the enterprise? How cost-effective?
Author Response
Thank you for your valuable comments on our study. We have carefully addressed each comment, revising our manuscript to enhance clarity as suggested. Please, see the attachment.

Reviewer 2 Report
Comments and Suggestions for Authors
In this article, a real-time monitoring system of small-scale multi-purpose juice extractor were investigated. These researches will promote the improvement of processing technology of the juicing processes. However, there are still some doubts in this work.
1、In Line 10,"According to the concept of smart postharvest management...", Should the concept of smart postharvest management be introduced in Introduction?
2、In line 13,Why was a smart monitoring system for optimizing the juicing processes in small- or medium-sized enterprises chosen for this study, not in large scale enterprises? Some explains should be provided in this article.
3、In line 37-38,In this sentence,“Juice extractors are the most widely used traditional food processing machines in food engineering. ”, should “...the most...”be revised? This sentence should be polished.
4、In line 48-49, “This is despite the growing emphasis on maximizing yield and profits in smart food processing [18–21]. ”, should “...despite...”be revised? This sentence should be polished.
5、In line 295-296, Should the contents and the textures in the raw materials and the processing materials of the pears and carrots be measured following the real-time measurement data to analysis these conclusions in this article, for example “...carrots have densely packed cells per unit area, resulting in a narrow fiber spacing and consequent hardness...”?
Author Response

(The authors gave the same response as above.)

Reviewer 3 Report
Comments and Suggestions for Authors
This manuscript proposed to implement a smart monitoring juicing system for real-time monitoring and optimizing the juicing process. However, the entire work reported in the manuscript was only recording the processing information collected from the equipment, without any 'smart' or 'optimization' included. The flowchart in Figure 2 showed that the project was only running a scenario where the variables were swept but not optimized. Following the scenario from this work, the real-time collected data needs to be analyzed after the juicing process for identifying the optimal condition/parameters, there would be no way to realize 'real-time optimization'. Such off-line optimization approach cannot be considered as a 'smart' system, hence i do not think the author accomplished what was proposed in the Introduction section, and this work cannot be accepted for publication.
Author Response

(The authors gave the same response as above.)

Round 2
Reviewer 1 Report
Comments and Suggestions for Authors
The revised-manuscript can be accepted after minor revisions.
The authors are encouraged to discuss the limitations of the study within the conclusion.